# ESPFormer: Doubly-Stochastic Attention with Expected Sliced Transport Plans

**Ashkan Shahbazi** [1]  **Elaheh Akbari** [1]  **Darian Salehi** [2]  **Xinran Liu** [1]  **Navid NaderiAlizadeh** [3]  **Soheil Kolouri** [1 4]

## Abstract

While self-attention has been instrumental in the success of Transformers, it can lead to over-concentration on a few tokens during training, resulting in suboptimal information flow. Enforcing doubly-stochastic constraints in attention matrices has been shown to improve structure and balance in attention distributions. However, existing methods rely on iterative Sinkhorn normalization, which is computationally costly. In this paper, we introduce a novel, fully parallelizable doubly-stochastic attention mechanism based on sliced optimal transport, leveraging Expected Sliced Transport Plans (ESP). Unlike prior approaches, our method enforces doubly stochasticity without iterative Sinkhorn normalization, significantly enhancing efficiency. To ensure differentiability, we incorporate a temperature-based soft sorting technique, enabling seamless integration into deep learning models. Experiments across multiple benchmark datasets, including image classification, point cloud classification, sentiment analysis, and neural machine translation, demonstrate that our enhanced attention regularization consistently improves performance across diverse applications. Our implementation code can be found at `https://github.com/dariansal/ESPFormer`.

## 1. Introduction

The debut of Transformers (Vaswani et al., 2017) marked a turning point in artificial intelligence and machine learning. Self-attention mechanisms excel at modeling the interactions among features, allowing Transformers to generate

highly expressive and context-rich representations that accurately capture the essence of input data (Khan et al., 2022). Although first developed to perform Natural Language Processing (NLP) tasks, the Transformer architecture has been adapted to a wide range of domains such as computer vision (Dosovitskiy et al., 2021; Touvron et al., 2021a;b; 2022), graphs (Yun et al., 2019; Rampášek et al., 2022; Shirzad et al., 2023), point clouds (Zhao et al., 2021; Qin et al., 2022), and biological sequences (Jumper et al., 2021; Abramson et al., 2024; Rives et al., 2021; Lin et al., 2023; Hayes et al., 2025). Over the years, a considerable amount of research has been devoted to enhancing the classic Transformer, focusing on better positional encoding (Dwivedi et al., 2021; Ying et al., 2021), efficiency of the attention mechanisms (Wang et al., 2020; Choromanski et al., 2020), and variants of self-attention (Hou et al., 2019; Ho et al., 2019), among others.

Self-attention produces a row-stochastic matrix, which can lead to a few tokens dominating the attention distribution. To mitigate this, enforcing doubly-stochastic attention ensures a more balanced distribution across tokens. To that end, Sander et al. (2022) introduced Sinkformer, replacing the softmax normalization in the classic Transformer with Sinkhorn's algorithm (Sinkhorn, 1964), resulting in a doubly-stochastic attention matrix. Sander et al. (2022) establish a connection between self-attention matrices and the optimal transport problem by theoretically demonstrating that Sinkformers can be interpreted as a Wasserstein gradient flow for an energy minimization in the infinite depth limit. Indeed, doubly stochastic attention can be interpreted as a transport plan between queries and keys, further strengthening the connection between optimal transport and attention mechanisms. Owing to their inherent doubly stochastic structure, transport plans (i.e., couplings between keys and queries) naturally serve as strong candidates for attention matrices in this setting.

Nevertheless, computing *optimal* transport plans between keys and queries is computationally expensive, with a general complexity of $\mathcal{O}(N^3)$ for $N$ tokens. A more scalable approach is entropy-regularized transport (Cuturi, 2013), which leverages the Sinkhorn algorithm to iteratively approximate the transport plan, obtaining a complexity $\mathcal{O}(SN^2)$, where $S$ denotes the number of iterations. The Sinkhorn algorithm (Sinkhorn, 1964), as used in Sinkform-

---

[1]Department of Computer Science, Vanderbilt University, Nashville, TN, USA. [2]Department of Computer Science, Duke University, Durham, NC, USA. [3]Department of Biostatistics and Bioinformatics, Duke University, Durham, NC, USA. [4]Department of Electrical and Computer Engineering, Vanderbilt University, Nashville, TN, USA. Correspondence to: Ashkan Shahbazi <ashkan.shahbazi@vanderbilt.edu>.

*Proceedings of the 42^nd International Conference on Machine Learning*, Vancouver, Canada. PMLR 267, 2025. Copyright 2025 by the author(s).

ers (Sander et al., 2022), repeatedly applies a series of row and column normalization steps until it reaches a desired level of convergence. Such an approach may introduce computational inefficiencies in scenarios where a significant number of iterations are needed for the normalization to converge.

A more computationally efficient alternative is the calculation of optimal transport plans for one-dimensional distributions, with a complexity of $\mathcal{O}(N \log N)$. This has motivated a large body of work on *sliced* optimal transport methods that compare distributions by comparing their slices, i.e., one-dimensional marginals (Rabin et al., 2012; Kolouri et al., 2016; 2018; 2019; Deshpande et al., 2019; Nguyen et al., 2022; Nguyen & Ho, 2024). Although sliced optimal transport methods offer efficient metrics between distributions, they do not explicitly construct transport plans. Recent studies have addressed this limitation by developing methods to construct transport plans through slicing (Mahey et al., 2024; Liu et al., 2025). Liu et al. (2025) introduce the Expected Sliced Transport Plan (ESP), which leverages slicing to define a computationally efficient metric while crucially providing an explicit transport plan by aggregating lifted plans from all slices. This makes ESP a strong candidate for doubly-stochastic attention mechanisms.

In this work, we leverage the ESP framework to propose a novel doubly-stochastic attention mechanism. We refer to the resulting architecture as ESPFormer. Our specific contributions are as follows:

- We propose ESPFormer, a novel doubly stochastic attention mechanism built on the recently introduced Expected Sliced Transport Plan (ESP) framework. ESP-Former ensures a more balanced distribution of attention across tokens while enabling control over the number of tokens each token attends to via an inverse temperature parameter.
- Through extensive experiments across diverse applications, we demonstrate performance improvements over both classic Transformer and Sinkformer architectures, along with enhanced computational efficiency compared to Sinkformer.
- We show that replacing the classic attention mechanism in a pre-trained Transformer with ESPFormer and fine-tuning for a few epochs results in significant performance gains.
- We show that by finetuning pretrained models with an exponential temperature annealing schedule and then switching to hard sorting at inference, we obtain exact doubly stochastic matrices reducing complexity from $O(N^2)$ to $O(LN \log N)$, and achieve consistent accuracy gains on the Cats vs. Dogs dataset.
- We demonstrate the compatibility of our proposed attention mechanism with the recently introduced differential attention architecture (Ye et al., 2025).

## 2. Background and Related Work

In this section, we first provide an overview of the existing variants of softmax attention, including the doubly-stochastic attention using Sinkhorn's algorithm (Sander et al., 2022). Then, we shift our focus to the fundamentals of sliced optimal transport, with soft sorting reviewed for algorithmic concerns. Finally, we provide an overview of Expected Sliced Transport Plans (Liu et al., 2025) and soft sorting (Prillo & Eisenschlos, 2020), which serve as the cornerstones of our proposed doubly-stochastic attention mechanism, ESPFormer.

### 2.1. Variants of Softmax Attention

At the heart of the Transformer architecture is the self-attention operation, a crucial component that enables dynamic pairwise interactions among tokens. In essence, it allows each position to "attend" to all others, with the degree of attention determined by how similar their representations are. Formally, let $W_Q, W_K \in \mathbb{R}^{m \times d}, W_V \in \mathbb{R}^{d \times d}$ denote the query, key, and value matrices, respectively. Then, for a sequence $(x_1, x_2, \cdots, x_N), x_i \in \mathbb{R}^d, \forall i$, the output of the attention function for the $i$-th row, $x_i$, can be written as

$$\frac{\sum_{j=1}^{N} \text{sim}(W_Q x_i, W_K x_j) W_V x_j}{\sum_{j=1}^{N} \text{sim}(W_Q x_i, W_K x_j)} \quad (1)$$

where $\text{sim}(\cdot, \cdot)$ can be any similarity function, and the normalization for the similarities is applied row-wise. The classic self-attention mechanism (Vaswani et al., 2017) leverages the softmax function to perform this row-wise normalization, i.e., the softmax of attention matrix $C$ with $C_{i,j} = (W_Q x_i)^T W_K x_j$ can be interpreted as row-wise normalization of $\exp(C)$. Alternative normalization operators have also been proposed in the literature. Some focus on normalizations that share the same properties as softmax but produce sparse outputs, such as SparseMax (Martins & Astudillo, 2016) and SparseK (Lou et al., 2024). Sparse-Max seeks the Euclidean projection of the input into the probabilistic simplex. Since this projection tends to hit the boundary of the simplex, SparseMax will output sparse probabilities. SparseK extends SparseMax by replacing the probabilistic simplex with a $k$-sum constraint. Others use different $\text{sim}(\cdot, \cdot)$ functions in (1) to achieve linear complexity: Katharopoulos et al. (2020) and Han et al. (2023) use kernel smoothers to linearize the similarity calculations, and Li et al. (2020) propose to approximate the exponential function in softmax by the first-order Taylor expansion.

Another line of research approaches the attention from the alignment/matching perspective and specifically utilizes concepts from the Optimal Transport (OT) theory. Zhang et al. (2021) propose to enforce the alignment between the distributions of keys and queries by adding a penalty term in the training objective, where the JensenShannon (JS)

divergence, the Wasserstein distance, and bi-directional conditional transport (Zheng & Zhou, 2021) are considered to define the matching cost. Xu & Chen (2023) introduce a multimodal co-attention method that relies on OT to match patches of images and gene embeddings for the survival prediction task. Zhang et al. (2023) reinterpret slot attention within the OT framework and propose to enhance the performance of slot attention by minimizing the entropy of the Sinkhorn divergence between two multisets, one containing inputs and the other containing slot features.

## 2.2. Doubly-Stochastic Attention

In their pioneering work, Sander et al. (2022) empirically observed that, during training, the row-stochastic attention matrices in classical Transformers tend to become approximately doubly-stochastic, with most column sums approaching 1. This suggests that Transformers inherently learn to distribute attention more evenly across tokens. In light of this finding, the authors propose the Sinkformer architecture, which replaces the softmax operation by the Sinkhorn's algorithm (Sinkhorn, 1964; Cuturi, 2013; Peyré et al., 2019) to enforce doubly-stochastic attention as an informative prior. They show that although both classic Transformers and Sinkformers can be understood as models that operate on discrete probability distributions, the Sinkformers have a special property that, in the infinite depth limit, they behave as a Wasserstein gradient flow for an energy minimization.

## 2.3. Sliced Optimal Transport

Consider the space of Borel probability measures on $\Omega \subset \mathbb{R}^d$ with finite $p$-th moment ($p \geq 1$), denoted by $\mathcal{P}_p(\Omega)$. For $\mu^1, \mu^2 \in \mathcal{P}_p(\Omega)$, the classic optimal transport (in the Kantorovich formulation) solves the optimization problem

$$\inf_{\gamma \in \Gamma(\mu_1, \mu_2)} \int_{\Omega^2} c(x, y) \mathrm{d}\gamma(x, y), \quad (2)$$

where $\Gamma(\mu^1, \mu^2)$ denotes the set of all couplings between $\mu^1$ and $\mu^2$, and $c : \Omega^2 \to \mathbb{R}_+$ is a lower semi-continuous function representing the transportation cost. Specifically, when $c(\cdot, \cdot)$ is the $p$-th power of a metric, the $p$-Wasserstein distance is defined as

$$W_p(\mu^1, \mu^2) \coloneqq \min_{\gamma \in \Gamma(\mu^1, \mu^2)} \left( \int_{\Omega^2} \|x - y\|^p \mathrm{d}\gamma(x, y) \right)^{\frac{1}{p}}. \quad (3)$$

In practical settings, the Wasserstein distance between discrete probability measures can be obtained by solving a linear program (Peyré et al., 2019) with an expensive computational complexity of $\mathcal{O}(N^3 \log N)$ for sample size $N$. Therefore, faster alternatives have been extensively studied, including entropy-regularized optimal transport (Cuturi, 2013) and sliced optimal transport (Rabin et al., 2012;

Kolouri et al., 2016; 2018; Deshpande et al., 2019; Kolouri et al., 2019; Nguyen et al., 2022; Nguyen & Ho, 2024). Sinkhorn's algorithm at the core of Sinkformer is used to solve the entropy-regularized optimal transport problem with an improved complexity of $\mathcal{O}(N^2 \log N)$.

Sliced optimal transport operates by projecting high-dimensional distributions onto one-dimensional slices, leveraging the key property that, in the one-dimensional case, a unique optimal transport plan exists, and the $p$-Wasserstein distance has a closed-form solution (Rabin et al., 2012),

$$W_p(\mu^1, \mu^2) = \left( \int_0^1 \left| F_{\mu^1}^{-1}(u) - F_{\mu^2}^{-1}(u) \right|^p \mathrm{d}u \right)^{\frac{1}{p}}, \quad (4)$$

where $F_{\mu^1}^{-1}$ and $F_{\mu^2}^{-1}$ are the quantile functions of $\mu^1$ and $\mu^2$ respectively, and the optimal transport map is given by $T(x) = F_{\mu^2}^{-1}(F_{\mu^1}(x))$ when $\mu^1$ is absolutely continuous. For empirical measures with $N$ samples, the quantile functions correspond to the sorted samples that can be calculated in $\mathcal{O}(N \log N)$. Then, by integrating the 1-dimensional Wasserstein distance over a set of $L$ slices, the sliced Wasserstein distance reduces the computational cost significantly to $\mathcal{O}(LN \log N)$.

## 2.4. Expected Sliced Transport Plans

Although the sliced Wasserstein distance offers a rapid and well-defined metric, it has one limitation: it does not generate a transport plan between the probability measures. It thus fails to explicitly provide how one distribution could be transported into another.

Liu et al. (2025) recently proposed the Expected Sliced Transport Plan (ESP), which defines an efficient transport plan as an aggregation of lifted optimal transport plans on 1-dimensional slices. Let $\mu^1 = \sum_{x \in \mathbb{R}^d} p(x) \delta_x$ be a discrete probability measure in $\mathcal{P}(\mathbb{R}^d)$, that is, $p(x) \geq 0$ for all $x \in \mathbb{R}^d$ and $\sum_{x \in \mathbb{R}^d} p(x) = 1$. We further assume that $p(x) \neq 0$ for at most countable many points $x \in \mathbb{R}^d$. Similarly, let $\mu^2 = \sum_{y \in \mathbb{R}^d} q(y) \delta_y$ be another probability measure in $\mathcal{P}(\mathbb{R}^d)$ with at most countable support. For a given $\theta \in \mathbb{S}^{d-1}$, the projected measures $\theta_{\#} \mu^1$ and $\theta_{\#} \mu^2$ are 1-dimensional probability measures in $\mathcal{P}(\mathbb{R})$, and there exists a unique optimal transport plan $\Lambda_\theta^{\mu^1, \mu^2}$ between them. Equivalently, $\theta_{\#} \mu^1$ and $\theta_{\#} \mu^2$ can be interpreted as probability measures over a quotient space $\mathbb{R}^d / \sim_\theta$, where $\sim_\theta$ is defined as follows:

$$x \sim_\theta x' \quad \text{if and only if} \quad \theta \cdot x = \theta \cdot x',$$

as each point on the slice $\mathbb{R}$ corresponds to an equivalent class of points in $\mathbb{R}^d$ that gets mapped to it by $\theta$. With a slight abuse of notation, we denote the equivalent class of $x \in \mathbb{R}^d$ by $\bar{x}^\theta$, referring to either a point in the quotient space $\mathbb{R}^d / \sim_\theta$ or the set of points $\{x' \in$

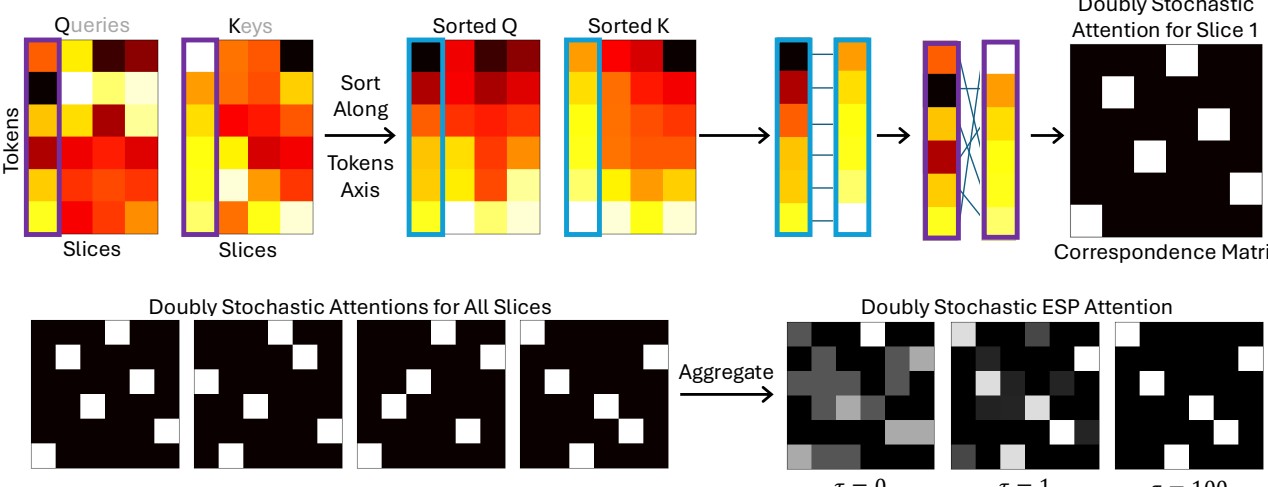

*Figure 1.* An overview of the proposed ESP attention mechanism. By integrating the slicing operator into the key and query matrices, each dimension is treated as a learnable slice. For each slice, tokens are (soft) sorted, and a doubly-stochastic correspondence matrix is computed between the keys and queries. Finally, these correspondence matrices across all dimensions are aggregated to form a single doubly-stochastic attention matrix.

$\mathbb{R}^d : \theta \cdot x' = \theta \cdot x\}$ interchangeably. Then, we can write $\theta_{\#}\mu^1 = \sum_{\bar{x}^\theta \in \mathbb{R}/\sim_\theta} P(\bar{x}^\theta)\delta_{\bar{x}^\theta}$, where $P(\bar{x}^\theta) = \sum_{x' \in \bar{x}^\theta} p(x')$, and $\theta_{\#}\mu^2 = \sum_{\bar{y}^\theta \in \mathbb{R}/\sim_\theta} Q(\bar{y}^\theta)\delta_{\bar{y}^\theta}$, where $Q(\bar{y}^\theta) = \sum_{y' \in \bar{y}^\theta} q(y')$. This quotient space interpretation of the 1-dimensional distributions $\theta_{\#}\mu^1$ and $\theta_{\#}\mu^2$ allows us to construct a lifted transport plan in the original space $\mathbb{R}^d$ using the optimal transport plan $\Lambda_\theta^{\mu^1,\mu^2}$,

$$\gamma_\theta^{\mu^1,\mu^2} := \sum_{x \in \mathbb{R}^d} \sum_{y \in \mathbb{R}^d} u_\theta^{\mu^1,\mu^2}(x,y)\delta(x,y), \qquad (5)$$

where the transported mass $u_\theta^{\mu^1,\mu^2}$ is defined as

$$u_\theta^{\mu^1,\mu^2}(x,y) := \frac{p(x)q(y)}{P(\bar{x}^\theta)Q(\bar{y}^\theta)}\Lambda_\theta^{\mu^1,\mu^2}(\{(\bar{x}^\theta, \bar{y}^\theta)\}).$$

Then for a given distribution of slicing directions ($\theta$'s): $\sigma \in \mathcal{P}(\mathbb{S}^{d-1})$, the ESP $\bar{\gamma}^{\mu^1,\mu^2} \in \Gamma(\mu^1, \mu^2)$ is defined as an expectation of $\gamma_\theta^{\mu^1,\mu^2}$ over $\sigma$:

$$\bar{\gamma}^{\mu^1,\mu^2} := \mathbb{E}_{\theta \sim \sigma}[\gamma_\theta^{\mu^1,\mu^2}] \qquad (6)$$

i.e. $\bar{\gamma}^{\mu^1,\mu^2}(\{(x,y)\}) = \int_{\mathbb{S}^{d-1}} \gamma_\theta^{\mu^1,\mu^2}(\{(x,y)\})d\sigma(\theta).$

Liu et al. (2025) have shown that the associated cost,

$$\mathcal{D}_p(\mu^1, \mu^2) = \left( \sum_{x \in \mathbb{R}^d} \sum_{y \in \mathbb{R}^d} \|x - y\|^p \bar{\gamma}^{\mu^1,\mu^2}(\{(x,y)\}) \right)^{\frac{1}{p}},$$

is a well-defined distance and equivalent to the Wasserstein distance.

## 2.5. Soft Sorting

Calculating the sliced Wasserstein distance involves evaluating the quantile functions of the distributions, which, in the discrete case, can be boiled down to the sorting operation. Sorting is one of the most common operations in computer science. Yet, the piecewise-linear sorted value function and the integer-valued rank/argsort operators pose a significant obstacle for gradient-based optimization techniques, which are essential in deep learning, as neither of them is differentiable. To incorporate sorting operations into the backpropagation framework, differentiable approximations, known as soft sorting, have been explored. Examples include smoothed rank operators by adding Gaussian noise (Taylor et al., 2008) and by using sigmoid surrogate functions (Qin et al., 2010), parameterizing permutations in terms of a differentiable relaxation (Mena et al., 2018), and relaxing the permutation matrices to be only row-wise stochastic (Grover et al., 2019). Of note, Cuturi et al. (2019) propose a differentiable proxy by viewing sorting as an optimal assignment problem and relaxing it to an entropic optimal transport problem from the input values to an auxiliary probability measure supported on an increasing family of target values.

Prillo & Eisenschlos (2020) introduce a simple yet highly effective continuous relaxation of the argsort operator using the softmax function. Given a vector $v \in \mathbb{R}^N$,

$$\text{SoftSort}_t^d(v) := \text{softmax}\left( \frac{-d(\text{sort}(v)\mathbb{1}^T, \mathbb{1}v^T)}{t} \right), \quad (7)$$

approximates the sorting permutation matrix, where the softmax operator is applied row-wise, $d(\cdot, \cdot)$ can be any differentiable semi-metric, and $t$ is a temperature parameter that controls the degree of the approximation.

# 3. Method

## 3.1. ESP for Uniform Discrete Distributions

Given our application of interest, we specifically focus on uniformly distributed discrete distributions with an equal number of support points, while the method can be readily extended to an unequal number of support points. For a given $N \in \mathbb{N}$, denote the space of uniform discrete distributions with $N$ support as

$$\mathcal{P}_{(N)}(\mathbb{R}^d) := \left\{ \frac{1}{N} \sum_{i=1}^{N} \delta_{x_i} | x_i \in \mathbb{R}^d, \forall i \in \{1, \cdots, N\} \right\}.$$

Let $\mu^1 = \frac{1}{N} \sum_{i=1}^{N} \delta_{x_i} \in \mathcal{P}_{(N)}(\mathbb{R}^d), \mu^2 = \frac{1}{N} \sum_{j=1}^{N} \delta_{y_j} \in \mathcal{P}_{(N)}(\mathbb{R}^d)$ and $X = [x_1, x_2, \cdots, x_N]^T \in \mathbb{R}^{N \times d}, Y = [y_1, y_2, \cdots, y_N]^T \in \mathbb{R}^{N \times d}$ be the corresponding matrix forms. Denote the symmetry group of order $N$ in matrix representation as $\mathbf{S}_N$, that is, $\mathbf{S}_N$ contains all permutation matrices of a set of $N$ elements.

Consider the 1-dimensional slice of $\mu^1$ and $\mu^2$ in the $\theta$ direction: $\theta_\# \mu^1 = \frac{1}{N} \sum_{i=1}^{N} \delta_{\theta \cdot x_i}$ and $\theta_\# \mu^2 = \frac{1}{N} \sum_{j=1}^{N} \delta_{\theta \cdot y_j}$, with $X\theta = [\theta \cdot x_1, \theta \cdot x_2, \cdots, \theta \cdot x_N]^T \in \mathbb{R}^N$ and $Y\theta = [\theta \cdot y_1, \theta \cdot y_2, \cdots, \theta \cdot y_N]^T \in \mathbb{R}^N$. There exists permutation matrices $A, B \in \mathbf{S}_N$ such that $AX\theta \in \mathbb{R}^N$ and $BY\theta \in \mathbb{R}^N$ are in sorted order, i.e.,

$$(AX\theta)_1 \le (AX\theta)_2 \le \cdots \le (AX\theta)_N;$$
$$(BY\theta)_1 \le (BY\theta)_2 \le \cdots \le (BY\theta)_N.$$

Then the optimal matching between $\theta_\# \mu^1$ and $\theta_\# \mu^2$ can be described by

$$(AX\theta)_n \mapsto (BY\theta)_n, \quad \forall n \in [1, 2, \cdots, N],$$

or equivalently, given that $A^T = A^{-1}$, $A^T B$ represents the transport map from $X\theta$ to $Y\theta$, i.e.,

$$(X\theta)_n \mapsto (A^T BY\theta)_n, \quad \forall n \in [1, 2, \cdots, N].$$

By lifting this transport map from the $\theta$ slice to the original space $\mathbb{R}^d$, we have $U_\theta := \frac{1}{N} A^T B$ which represents a transport plan from $X$ to $Y$.

Finally, for a given histogram of $\theta$, $\sigma = \sum_{l=1}^{L} \sigma_l \delta_{\theta_l}$ with $\sum_{l=1}^{L} \sigma_l = 1$, the ESP for uniformly-distributed discrete distributions with the same number of supports is defined as

$$G := \sum_{l=1}^{L} \sigma_l^\tau U_{\theta_l}, \tag{8}$$

where $\tau$ denotes the "inverse temperature" hyperparameter, and $\sigma_l^\tau$ is defined as

$$\sigma_l^\tau = \frac{e^{-\tau \mathcal{D}_p^p(\mu^1, \mu^2; \theta_l)}}{\sum_{l'=1}^{L} e^{-\tau \mathcal{D}_p^p(\mu^1, \mu^2; \theta_{l'})}}, \tag{9}$$

with $\mathcal{D}_p^p(\mu^1, \mu^2; \theta)$ representing the $p$-th power of the induced transport cost by $U_\theta$, defined as

$$\mathcal{D}_p^p(\mu^1, \mu^2; \theta) = \sum_{i=1}^{N} \sum_{j=1}^{N} \|x_i - y_j\|^p U_\theta(\{(x_i, y_j)\}), \tag{10}$$

where $U_\theta(\{(x_i, y_j)\})$ is the $(i, j)'$th component of the $U_\theta$ matrix. When $\tau = 0$, $\sigma_l = \frac{1}{L}$ for all $l \in [1, 2, \cdots, L]$, then $G$ is simply the mean of all the lifted plans.

When handling uniformly distributed discrete distributions with different numbers of support points (e.g., in cross-attention), the transport plan involving mass splitting can be approximated using linear interpolation. For $\mu^1 = \frac{1}{N} \sum_{j=1}^{N} \delta_{x_i}, \mu^2 = \frac{1}{M} \sum_{j=1}^{M} \delta_{y_j}$ with $N \neq M$, let $A' \in \mathbb{R}^{N \times N}$ and $B' \in \mathbb{R}^{M \times M}$ respectively denote their sorting permutation matrices. We define an interpolation matrix $I \in \mathbb{R}^{N \times M}$, where

$$I[i, j] := \begin{cases} \frac{\frac{i}{N} - \frac{j}{M}}{\frac{1}{M}}, & \text{if } \frac{j}{M} \le \frac{i}{N} \le \frac{j+1}{M}; \\ \frac{\frac{j}{M} - \frac{i}{N}}{\frac{1}{M}}, & \text{if } \frac{j-1}{M} \le \frac{i}{N} \le \frac{j}{M}; \\ 0, & \text{elsewhere.} \end{cases} \tag{11}$$

The transport plan is then given by $U_\theta = \frac{1}{N} A'^T I B'$.

## 3.2. ESP Attention

To bridge the abstract formulation of the Expected Sliced Plan (ESP) with its practical use in attention mechanisms, we interpret the query and key matrices as empirical probability measures due to their permutation-equivariant nature. Following our notation in Section 2.1, let $W_Q, W_K \in \mathbb{R}^{m \times d}, W_V \in \mathbb{R}^{d \times d}$ denote the query, key, and value matrices, respectively, and let $X = [x_1, x_2, \cdots, x_N], x_i \in \mathbb{R}^d, \forall i$ denote the input tokens. Also, let $Q = W_Q X \in \mathbb{R}^{m \times N}$ and $K = W_K X \in \mathbb{R}^{m \times N}$ denote the queries and keys, respectively. Then, we think of doubly-stochastic attention as a transport plan between the empirical measures $\mu^Q$ and $\mu^K$, defined as

$$\mu^Q = \frac{1}{N} \sum_{i=1}^{N} \delta_{q_i}, \quad \mu^K = \frac{1}{N} \sum_{j=1}^{N} \delta_{k_j}.$$

To create such a transport plan using the ESP framework presented in Section 3.1, we use a set of $L$ slicing directions $\Theta = [\theta_1, ..., \theta_L]^T \in \mathbb{R}^{L \times m}, \theta_l \in \mathbb{S}^{m-1}$. Then, the rows of $\Theta K$ and $\Theta Q$ contain the projected keys and queries for each slice. The transportation plan $U_{\theta_l}$ is, then, calculated for each slice by soft-sorting the projected keys and queries as described in Section 3.1. Finally, the attention matrix is calculated from (8).

A key consideration is the choice of slices, $\Theta$. While classic sliced OT uniformly samples slices from $\mathbb{S}^{m-1}$, prior work suggests learning them (Deshpande et al., 2019; Naderial-izadeh et al., 2021; NaderiAlizadeh et al., 2025). Though

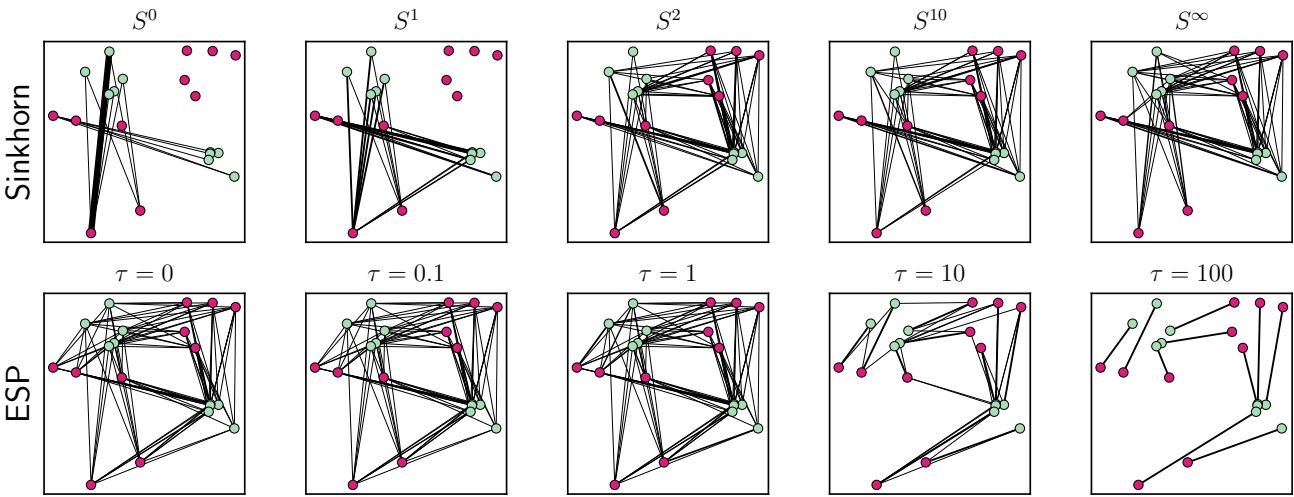

*Figure 2.* The attention weights between an example pair of keys (red) and queries (green) obtained by Sinkhorn's algorithm (top row) with different numbers of iterations and by Expected Sliced Transport Plans (bottom row) with different inverse temperature values. Note that for Sinkrhorn and at zero iterations, i.e., $S^0$, the computed attention reduces to classic self-attention. The weights are represented by the width of the lines connecting each pair of points.

this approach is common, learning $\Theta$ introduces additional parameters, increasing the total count in ESP attention and complicating fair comparisons with other mechanisms. Notably, since the input distributions (keys and queries) are themselves learned, optimizing $\Theta$ may be unnecessary. Instead, the distributions can adapt to a fixed slicing scheme, as done in prior work, such as FSPool (Zhang et al., 2020). To avoid extra parameters, in this work, we propose using axis-aligned slices by setting $\Theta = \mathbb{I}_{m \times m}$, the identity matrix. This leads to

$$\text{ESP-Attention}(Q, K, V) = VG, \qquad (12)$$

where $V = W_V X \in \mathbb{R}^{d \times N}$ is the value matrix. Figure 1 illustrates the construction of the ESP matrix $G$, which serves as the proposed transport-based attention map, directing how information flows from $V$ to the output. Moreover, Figure 2 compares classic attention (Sinkhorn $S^0$), Sinkhorn attention from (Sander et al., 2022) across different iteration counts, and our ESP attention under varying "inverse temperature" hyperparameters. As can be seen, while ESP provides a balanced distribution of attention due to its double stochasticity, the "inverse temperature" parameter controls the number of other tokens each token should pay attention to. The overall pipeline of ESPFormer can be found in Algorithm 1. For completeness, we also report results from explicitly learning the slices and observe no significant benefits compared to axis-aligned slices.

### 3.3. Runtime Complexity

ESPFormer begins with query and key projections ($Q = W_Q X$ and $K = W_K X$), each requiring $\mathcal{O}(mNd)$ opera-

---

**Algorithm 1** ESPFormer's Doubly-Stochastic Attention

**input** Query matrix $Q \in \mathbb{R}^{m \times N}$, Key matrix $K \in \mathbb{R}^{m \times N}$, Value Matrix $V \in \mathbb{R}^{d \times N}$, SoftSort hyperparameter $t$, and "inverse temperature" hyperparameter $\tau$.

**output** Attention-weighted output matrix
1: Calculate the pairwise distance matrix:

$$[C]_{ij} = \|Q_{:i} - K_{:j}\|^2.$$

2: **for** $l = 1$ to $m$ **do**
3:      SoftSort the projected samples using (7):

$$A_l = \text{SoftSort}_t(Q_{l:}), B_l = \text{SoftSort}_t(K_{l:}).$$

4:      Calculate the transport plan $U_l = \frac{1}{N} A_l^T B_l$
5:      Calculate $D_l = \sum_{ij}[C]_{ij}[U_l]_{ij}$
6: **end for**
7: Calculate the $\sigma^\tau = \text{softmax}(D; \tau)$
8: Aggregate the plans from all slices $G = \sum_{l=1}^{m} \sigma_l^\tau U_l$
9: Return: $VG$

---

tions, where $d$ corresponds to the output dimension. The subsequent SoftSort operation, which involves computing pairwise distances, incurs a complexity of $\mathcal{O}(N^2)$ per slice. When applied across all $L = m$ slices, this yields a complexity of $\mathcal{O}(mN^2)$. The computation of transport plans through matrix multiplications with soft permutation matrices contributes an additional $\mathcal{O}(mN^2)$ operations. The final aggregation of transport plans requires summing $m$ plans of size $N \times N$, also contributing $\mathcal{O}(mN^2)$ operations. Therefore, the overall runtime complexity of ESPFormer is $\mathcal{O}(mN(N + d))$. In comparison, Sinkhorn's algorithm

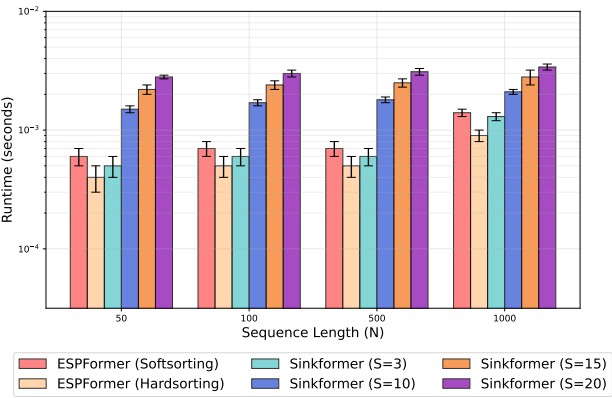

*Figure 3.* Runtime analysis of ESPFormer versus Sinkformer with varying iteration counts ($S$) for sequence lengths $N \in \{50, 100, 500, 1000\}$ with $d = 1024$, averaged over 10 runs. ESPFormer achieves superior computational efficiency across all sequence lengths with hard sorting, and outperforms Sinkformer for $S > 3$ under soft sorting, while maintaining model expressivity. .

exhibits a runtime complexity of $\mathcal{O}((S + m)N^2)$, where $S$ denotes the number of iterations. A key distinction lies in the parallelization capabilities: while Sinkhorn necessitates sequential processing over $S$ iterations, ESPFormer enables parallel computation across the $m$ slices. This parallelizability allows ESPFormer to scale efficiently with increasing $m$. As demonstrated in Figure 3, ESPFormer offers superior efficiency at all sequence lengths with hard sorting, and surpasses Sinkformer for $S > 3$ when using soft sorting. The wall clock runtime analysis of ESPFormer compared to all the baselines can be found in Appendix A.

## 4. Experiments

In this section, we evaluate the performance of ESPFormer across a diverse set of domains, including image classification using ViTs (Dosovitskiy et al., 2021), point cloud classification using set transformers (Lee et al., 2019) and point cloud transformers (Guo et al., 2021), as well as sentiment analysis and neural machine translation via Transformers, highlighting the applicability of our approach in comparison to Differential Transformer (Ye et al., 2025), Sinkformer (Sander et al., 2022), and the Vanilla Transformer (Vaswani et al., 2017). Throughout our experiments, we employed an axis-aligned slicer. our experimental setup and implementation can be found in Appendices B and C.

### 4.1. ModelNet40 Classification

The ModelNet40 dataset (Wu et al., 2015) comprises 40 widely recognized 3D object categories and serves as a standard benchmark for point cloud classification. Transformers designed for point clouds and sets have been extensively evaluated on ModelNet40, with notable examples including Set Transformers (Lee et al., 2019) and Point Cloud Trans-

formers (Guo et al., 2021). Table 2 presents the classification results across four runs, comparing different attention mechanisms integrated into Set Transformers and Point Cloud Transformers. Notably, ESPFormer outperforms competing methods, demonstrating superior performance.

### 4.2. Sentiment Analysis

We next evaluate ESPFormer on the IMDB dataset (Maas et al., 2011) for sentiment analysis. Following Sander et al. (2022), our architecture comprises an attentionbased encoder followed by a maxpooling layer trained to classify movie reviews as either positive or negative. Table 3 presents the accuracy improvements achieved by using ESP-Former as the core attention module, compared to baseline models, showcasing its robustness in text classification.

We additionally evaluate ESPFormer on the TweetEval sentiment dataset (Barbieri et al., 2020), where tweets are labeled positive or negative. We use the same encoder + maxpooling architecture and training setup as for IMDb. Table 4 reports the corresponding accuracy metrics.

### 4.3. Neural Machine Translation

We evaluate ESPFormer and Sinkformer using two reference models: the Transformer and its DiffTransformer counterpart (Ye et al., 2025). Both models are trained using the fairseq sequence modeling toolkit (Ott et al., 2019) on the IWSLT'14 German-to-English dataset (Cettolo et al., 2014). The architecture of both models consists of an encoder and a decoder, each with a depth of 6 layers. Initially, we trained both models for 25 epochs using the standard training procedure. After this phase, we performed a Plug-and-Play evaluation, where the attention heads were plugged into the pre-trained models and evaluated on their performance, as shown in Table 5. In this phase, we tested ESPFormer and Sinkformer attention heads, denoted by the respective models in the table, comparing their performance to the base Transformer and DiffTransformer models. Following the Plug-and-Play evaluation, we performed a Fine-Tune Boost phase, where the models were further fine-tuned for an additional 10 epochs. The fine-tuning led to further performance gains, as observed in the table, with ESPFormer showing the highest improvement in both the Transformer and DiffTransformer settings, achieving the best BLEU score of 34.64 and 34.83, respectively. The fine-tuned results show the effectiveness of incorporating ESPFormer into the model architecture.

### 4.4. Vision Transformers

To evaluate the generalizability of the models under limited data scenarios, we conducted experiments on the Cats and Dogs dataset (Kaggle, 2013) using varying fractions of the training data: 1%, 10%, 25%, and 100%. We train a

*Table 1.* Average and standard deviation (over 3 runs) of ESPFormer's classification accuracy (%) vs. baselines on the Cats and Dogs dataset under varying data availability. ESPFormer's performance is reported in three modes: initial soft sort, sharp soft sort, and hard sort.

| Data Fraction | Baselines | | | ESPFormer | | |
|---|---|---|---|---|---|---|
| | Sinkformer | DiffTransformer | Transformer | Initial Soft Sort | Sharp Soft Sort | Hard Sort |
| **1%** | $55.07 \pm 3.34$ | $53.78 \pm 0.28$ | $49.71 \pm 0.31$ | $55.66 \pm 3.95$ | $57.86 \pm 3.77$ | **$58.52 \pm 3.73$** |
| **10%** | $69.56 \pm 0.32$ | $67.34 \pm 0.11$ | $57.25 \pm 0.22$ | $71.49 \pm 0.43$ | $72.22 \pm 0.37$ | **$72.71 \pm 0.36$** |
| **25%** | $74.56 \pm 0.58$ | $74.86 \pm 0.17$ | $72.25 \pm 0.16$ | $75.40 \pm 0.38$ | $75.92 \pm 0.31$ | **$75.92 \pm 0.28$** |
| **100%** | $79.12 \pm 0.17$ | $78.85 \pm 0.11$ | $78.49 \pm 0.09$ | $79.47 \pm 0.12$ | $80.61 \pm 0.11$ | **$81.23 \pm 0.11$** |

*Table 2.* Test accuracy (%) on the ModelNet40 dataset over 4 runs. Accuracies marked with $*$ are reported from (Sander et al., 2022).

| Model | Best | Median | Mean | Worst |
|---|---|---|---|---|
| Set Transformer* | 87.8 | 86.3 | 85.8 | 84.7 |
| Set DiffTransformer | 89.0 | 88.7 | 88.7 | 88.6 |
| Set Sinkformer* | 89.1 | 88.4 | 88.3 | 88.1 |
| Set ESPFormer | **89.6** | **89.5** | **89.4** | **89.1** |
| Point Cloud Transformer* | **93.2** | 92.5 | 92.5 | 92.3 |
| Point Cloud DiffTransformer | 93.1 | 92.8 | **92.7** | **92.6** |
| Point Cloud Sinkformer* | 93.1 | 92.8 | **92.7** | 92.5 |
| Point Cloud ESPFormer | **93.2** | 92.9 | **92.7** | **92.6** |

*Table 3.* Test accuracy (%) for Sentiment Analysis on IMDb.

| Model | Best | Median | Mean | Worst |
|---|---|---|---|---|
| Transformer | 85.30 | 85.25 | 85.25 | 85.20 |
| DiffTransformer | **85.50** | 85.45 | 85.45 | **85.40** |
| Sinkformer | 85.40 | 85.39 | 85.37 | 85.30 |
| ESPFormer | **85.50** | **85.50** | **85.47** | **85.40** |

*Table 4.* Test accuracy (%) for Sentiment Analysis on TweetEval.

| Model | Best | Median | Mean | Worst |
|---|---|---|---|---|
| Transformer | 71.50 | 71.35 | 71.31 | 71.10 |
| DiffTransformer | **72.60** | 72.35 | 72.31 | 72.00 |
| Sinkformer | 72.40 | 72.30 | 72.23 | 71.90 |
| ESPFormer | **72.60** | **72.40** | **72.36** | **72.10** |

ViT and modify the attention mechanism accordingly. The models compared include ESPFormer, Sinkformer, Diff-Transformer, and the standard Transformer. The goal was to assess how well each model performs when faced with progressively larger amounts of data, particularly in resource-constrained settings. Table 1 summarizes the classification accuracy for each model under different data fractions. Our results demonstrate that ESPFormer consistently outperforms the other models across all data fractions, achieving the highest classification accuracy at each level of data availability. With only 1% of the data, ESPFormer achieves 55.66% accuracy, a significant improvement over the Transformer at 49.71%. As the data availability increases, ESP-Former continues to show superior performance. This suggests that ESPFormer is particularly robust in data-scarce environments, showcasing its ability to generalize well even under limited training data.

*Table 5.* Plug-and-Play and Fine-Tune Boost performance of Transformer and DiffTransformer Baselines on the IWSLT14 German-to-English dataset, reported as the median over 4 runs. Results with a $\star$ sign denote the Plug-and-Play performance of a different attention module than the base attention module.

| | Model | Plug-and-Play | Fine-Tune Boost |
|---|---|---|---|
| Transformer | Transformer | 33.40 | 34.61 |
| | Sinkformer | $33.36^{\star}$ | 34.61 |
| | ESPFormer | $33.38^{\star}$ | **34.64** |
| DiffTransformer | DiffTransformer | $33.85^{\star}$ | 34.78 |
| | Sinkformer | $33.67^{\star}$ | 34.81 |
| | ESPFormer | $33.72^{\star}$ | **34.83** |

### TRANSITION TO HARD SORTING

During training, the use of soft sorting—which depends on a temperature parameter—results in approximate doubly stochastic matrices (DSMs). However, this approximation can be addressed by gradually annealing the temperature during training, effectively transitioning toward hard sorting. Consequently, at inference time, soft sorting can be replaced with hard sorting to yield exact DSMs. This not only ensures exact computations but also reduces the inference time from $O(N^2)$ to $O(LN \log N)$, where $L$ is the number of slices.

To further clarify this point, we conducted additional experiments the Cats vs. Dogs dataset using the same fractions of the training data to evaluate the effect of hard sorting at inference time. We finetuned pretrained models for 40 epochs using an exponential temperature annealing schedule to progressively sharpen the sorting behavior. Specifically, the temperature at epoch $e$ is defined as: $\text{temp}(e) = \gamma \text{temp}(e-1)$, with $\gamma = 0.8$. This decay schedule reduces the temperature to approximately $10^{-6}$, making the soft sorting operation nearly equivalent to a hard permutation. After finetuning, we replaced soft sorting with hard sorting for final evaluation. Table 1 demonstrates that fine-tuning with temperature annealing and switching to hard sorting yields consistent improvements in test accuracy.

### ON THE CHOICE OF SLICER

We employ axis-aligned slicing because our keys and queries are learned parameters; any optimal slice orientation

*Table 6.* Accuracy (%) over three runs (mean $\pm$ standard deviation) across slicer types, slice counts $L$, and inverse temperature $\tau$.

| | $L = 1$ | $L = 8$ | $L = 32$ | $L = 64$ | $L = 128$ |
|---|---|---|---|---|---|
| | | | $\tau = 0.1$ | | |
| **Learnable** | $74.30 \pm 0.48$ | $78.70 \pm 0.32$ | $79.10 \pm 0.22$ | $78.40 \pm 0.26$ | $76.20 \pm 0.42$ |
| **Frozen** | $66.50 \pm 0.52$ | $72.80 \pm 0.38$ | $78.30 \pm 0.18$ | $79.20 \pm 0.30$ | $79.60 \pm 0.28$ |
| **Axis-Aligned** | – | – | – | $79.47 \pm 0.12$ | – |
| | | | $\tau = 1.0$ | | |
| **Learnable** | $74.30 \pm 0.48$ | $79.07 \pm 0.30$ | $78.20 \pm 0.35$ | $77.80 \pm 0.21$ | $74.10 \pm 0.46$ |
| **Frozen** | $66.50 \pm 0.52$ | $73.11 \pm 0.43$ | $77.95 \pm 0.25$ | $78.80 \pm 0.29$ | $78.40 \pm 0.27$ |
| **Axis-Aligned** | – | – | – | $78.85 \pm 0.31$ | – |
| | | | $\tau = 10$ | | |
| **Learnable** | $74.30 \pm 0.48$ | $79.15 \pm 0.29$ | $78.06 \pm 0.27$ | $77.10 \pm 0.23$ | $74.15 \pm 0.44$ |
| **Frozen** | $66.50 \pm 0.52$ | $73.45 \pm 0.41$ | $76.85 \pm 0.24$ | $77.85 \pm 0.30$ | $78.10 \pm 0.26$ |
| **Axis-Aligned** | – | – | – | $77.75 \pm 0.32$ | – |

is thus implicitly encoded in the projection matrices $W_Q$ and $W_K$. This avoids introducing extra learnable parameters and ensures a fair comparison with baselines such as vanilla attention and Sinkformer.

However, for an ablation study, we ran additional experiments on the Cats vs. Dogs dataset, varying the number of slices $L$, the inverse temperature $\tau$, and whether the slicer was learnable or frozen. Tables 6 summarizes the results. Across all settings, axis-aligned slicing provides a simple, interpretable baseline without parameter overhead. Learnable slicers can boost performance when $L$ is small, by focusing on the most informative directions, but their advantage diminishes as $L$ grows. Frozen slicers likewise require sufficiently many slices to approximate the distribution structure effectively, since performance degrades when $L$ is low in high-dimensional spaces.

## 5. Conclusion

We introduced ESPFormer, a novel Transformer architecture that integrates a fast, doubly-stochastic attention mechanism based on Expected Sliced Transport Plans (ESP). By leveraging differentiable soft sorting and a fully parallelizable approach, ESPFormer provides a computationally efficient alternative to iterative Sinkhorn normalization while ensuring a balanced distribution of attention across tokens. Our experiments across diverse domains, including image classification, point cloud processing, sentiment analysis, and neural machine translation, demonstrate that ESPFormer consistently outperforms both classical self-attention and Sinkhorn-based alternatives in terms of accuracy and efficiency. Furthermore, we showed that our proposed attention can be seamlessly integrated into pre-trained Transformers, improving performance even with minimal fine-tuning. The flexibility of ESPFormer also makes it compatible with emerging differential attention mechanisms, e.g., (Ye et al., 2025), expanding its applicability to a broad range of architectures. These findings highlight the potential of transport-

based attention mechanisms as a principled alternative to existing methods, paving the way for future research in efficient and structured attention models.

## Limitations

Our method has two main limitations. First, it is incompatible with strictly causal Transformer or autoregressive architectures, since enforcing both lower-triangularity and doubly stochasticity reduces the mapping to the identity. Second, during training its memory usage scales linearly with the number of slices, as we must retain each slices permutation matrix; after fine-tuning and switching to hard sorting, however, these matrices can be discarded and all inference computations can be run off-GPU.

## Impact Statement

ESPFormer significantly accelerates the computation of balanced attention by introducing an efficient mechanism for enforcing doubly stochastic constraints through the mathematics of sliced optimal transport. The proposed attention improves model robustness and interoperability, and is shown to be effective even in a plug-and-play setting. This enables practical use of balanced attention as a strong inductive bias, particularly effective in low-data regimes. Furthermore, by establishing a connection between sliced optimal transport and attention mechanisms, ESPFormer opens a new research direction for structured and efficient attention through transport-based formulations. Although ESPFormer has the potential to impact a broad range of applications, we do not foresee any specific societal risks requiring special attention.

## Acknowledgment

This work was partially supported by the NSF CAREER Award #2339898 and the Wellcome Leap 'Surgery: Assess/-Validate/Expand (SAVE)' program.

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

# A. Full runtime wall-clock analysis, and Patch size impact

## A.1. Full runtime wall-clock

The runtime comparison in Figure 4 highlights the computational efficiency of different attention mechanisms across varying sequence lengths. Transformer and DiffTransformer exhibit the lowest runtime due to their standard self-attention mechanism, making them computationally lightweight. In contrast, ESPFormer achieves a balance between efficiency and expressivity, maintaining lower runtimes compared to Sinkformer across all sequence lengths. These results emphasize the trade-off between computational efficiency and model expressivity when selecting an attention model for large-scale tasks.

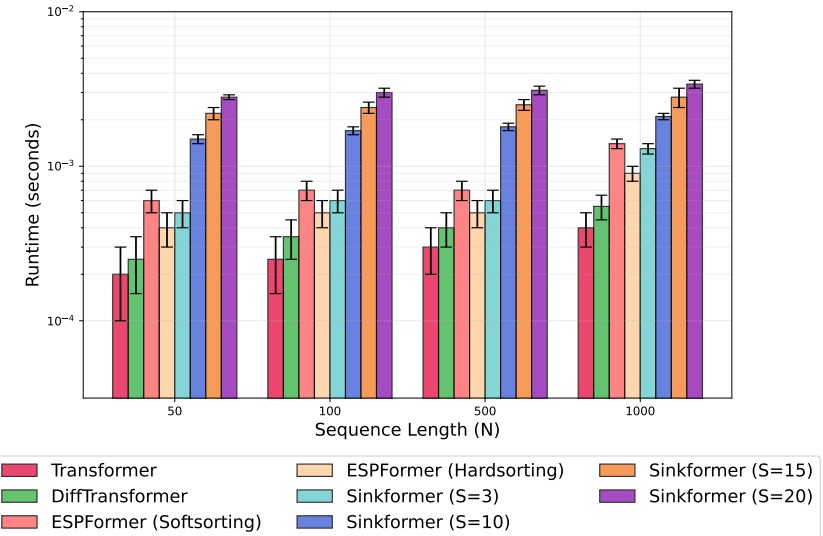

*Figure 4.* Runtime analysis of Transformer, DiffTransformer, ESPFormer, and Sinkformer with varying iteration counts ($S$) for sequence lengths $N \in \{50, 100, 500, 1000\}$ with $d = 1024$, averaged over 10 runs. ESPFormer consistently demonstrates superior computational efficiency compared to Sinkformer across all sequence lengths, while Transformer and DiffTransformer remain the most lightweight.

## A.2. Impact of Patch Size

Similar to Sander et al. (2022) and to analyze the effect of patch size on model performance, we trained Transformer, DiffTransformer, Sinkformer, and ESPFormer on the MNIST dataset (LeCun et al., 1998). To isolate the impact of the attention mechanism, we used a single-layer self-attention module without feed-forward layers. Figure 5 illustrates the test accuracy as a function of patch size for each model. ESPFormer consistently outperforms competing models, particularly for smaller patch sizes, highlighting its superior ability to capture fine-grained details. As the patch size increases, the performance gap narrows, with all models converging to similar accuracies — likely due to the reduced information content within each individual patch.

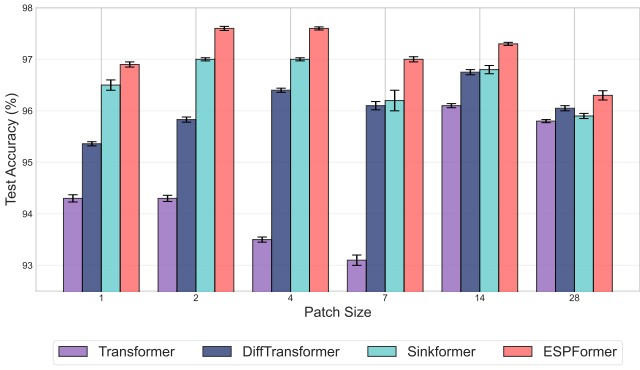

*Figure 5.* Comparison of MNIST test accuracy across different patch sizes for Transformer, DiffTransformer, Sinkformer, and ESPFormer architectures. Results are averaged over 3 runs.

# B. Implementation Details

### Sinkhorn's Algorithm

We implement Sinkhorns algorithm in the log domain to enhance numerical stability. Given a matrix $S^0 \in \mathbb{R}^{n \times n}$ defined as $S_{i,j}^0 = e^{C_{i,j}}$ for some cost matrix $C \in \mathbb{R}^{n \times n}$, the algorithm iteratively updates the scaling vectors $(f, g) \in \mathbb{R}^n \times \mathbb{R}^n$ such that the limiting matrix is given by

$$S^\infty = \text{diag}(e^{f^\infty}) S^0 \text{diag}(e^{g^\infty}).$$

Starting with $g^0 = 0_n$, the updates are performed in the log domain as follows:

$$f^{l+1} = \log(\mathbf{1}_n/n) - \log(S e^{g^l}), \quad \text{if } l \text{ is even,}$$

$$g^{l+1} = \log(\mathbf{1}_n/n) - \log(S^\top e^{f^l}), \quad \text{if } l \text{ is odd.}$$

This log-domain formulation allows for efficient and numerically stable computations by leveraging the `log-sum-exp` trick to evaluate expressions like $\log(S e^{g^l})$ and $\log(S^\top e^{f^l})$.

### Implementation of Sinkhorn Algorithm:

```python
def sinkhorn_log_domain(C, epsilon=1e-3, num_iters=50):
    n = C.shape[0]
    S = np.exp(-C / epsilon)  # Compute S^0
    f, g = np.zeros(n), np.zeros(n)  # Initialize log scaling vectors

    for l in range(num_iters):
        if l % 2 == 0:
            f = np.log(1 / n) - log_sum_exp(np.log(S) + g)
        else:
            g = np.log(1 / n) - log_sum_exp(np.log(S.T) + f)

    return f, g
```

For further details and implementation, please refer to the **SinkFormer GitHub Repository**.

### Differential Transformer

The Differential Transformer extends traditional self-attention by introducing differential attention mechanisms, which modulate the contribution of multiple attention maps using a learnable parameter $\lambda$. This allows the model to capture finer differences in the relational structure of the input data.

Given an input sequence $X$, the attention mechanism computes two different sets of queries and keys, denoted as $(Q_1, K_1)$ and $(Q_2, K_2)$. The attention output is computed using a weighted difference of the attention maps:

$$A = \text{softmax}(Q_1 K_1^T) - \lambda \times \text{softmax}(Q_2 K_2^T).$$

The resulting attention map is then applied to the values $V$, followed by normalization and projection:

$$O = \text{GroupNorm}(AV).$$

### Implementation of Differential Attention:

```python
def DiffAttn(X, W_q, W_k, W_v, \lambda):
    Q1, Q2 = split(X @ W_q)
```

```
    K1, K2 = split(X @ W_k)
    V = X @ W_v
    s = 1 / sqrt(d)
    A1 = Q1 @ K1.transpose(-1, -2) * s
    A2 = Q2 @ K2.transpose(-1, -2) * s
    return (softmax(A1) - \lambda * softmax(A2)) @ V
```

**Implementation of Multi-Head Differential Attention:**

```
def MultiHead(X, W_q, W_k, W_v, W_o, \lambda):
    O = GroupNorm([DiffAttn(X, W_qi, W_ki, W_vi, \lambda) for i in range(h)])
    O = O * (1 - \lambda{init})
    return Concat(O) @ W_o
```

For further details and implementation, please refer to the **Differential Transformer Github Repository.**

## C. Experiment Details

### C.1. Table of Notations

Table 7 includes the notations used in this section.

*Table 7.* Experimental Notations

| Notation | Description |
|---|---|
| t | Temperature parameter in the Soft Sorting algorithm |
| $\tau$ | The "inverse temperature" hyperparameter in ESP |
| $\epsilon$ | The entropy regularization parameter in the Sinkhorn algorithm |
| $\mathcal{S}$ | Number of iterations in Sinkformer |
| $\lambda_{\text{initial}}$ | Initial $\lambda$ value in DiffTransformer |
| $lr_{initial}$ | Initial learning rate for all methods |
| $I$ | The interpolation matrix in ESPformer |

### C.2. ModelNet 40

In our experiments on ModelNet40 utilizing Set Transformers, we begin by preprocessing the dataset and uniformly sampling 5000 points from each object.

**Set Transformers**
The model architecture consists of two Induced Set Attention Blocks (ISAB) in the encoder, followed by a decoder incorporating a Set Attention Block (SAB) and a Pooling by Multihead Attention (PMA) module. The training procedure employs a batch size of 64 and utilizes the **Adam optimizer** (Kingma & Ba, 2015). The network is trained for 300 epochs, with an initial learning rate of $10^{-3}$, which is reduced by a factor of 10 after 200 epochs.

| t | $\tau$ | $\epsilon$ | $\mathcal{S}$ | $\lambda_{\text{initial}}$ | $lr_{initial}$ | $I$ |
|---|---|---|---|---|---|---|
| $10^{-3}$ | $10^{-1}$ | $10^{-1}$ | 21 | 0.8 | $10^{-3}$ | $Identity_{N \times M}$ |

*Table 8.* Hyperparameters used in training for Set Transformers on ModelNet40 dataset.

**Point Cloud Transformers**
The training is conducted with a batch size of 32 using Stochastic Gradient Descent (SGD) (Ruder, 2016). The model undergoes 300 training epochs, starting with an initial learning rate of $10^{-4}$, which is reduced by a factor of 10 after 250 epochs.

| t | $\tau$ | $\epsilon$ | $\mathcal{S}$ | $\lambda_{\text{initial}}$ | $lr_{initial}$ | $I$ |
|---|---|---|---|---|---|---|
| $10^{-3}$ | $10^{-1}$ | $10^{-1}$ | 21 | 0.8 | $10^{-4}$ | $Identity_{N \times N}$ |

*Table 9.* Hyperparameters used in training for Point Cloud Transformers on ModelNet40 dataset.

### C.3. Sentiment Analysis

For our sentiment analysis experiments, we utilize the publicly available implementation from the `nlp-tutorial`[1] repository, where a pretrained Transformer model is fine-tuned on the IMDb dataset. In our experimental setup, we reset the parameters of the pretrained Transformer and train it from scratch on the IMDb dataset. The model architecture consists of a depth of 6 layers and employs 8 attention heads. Training is conducted using the Adam optimizer with a batch size of 32 over 15 epochs. The initial learning rate is set to $10^{-4}$ and is reduced by a factor of 10 after 12 epochs.

| t | $\tau$ | $\epsilon$ | $\mathcal{S}$ | $\lambda_{\text{initial}}$ | $lr_{initial}$ | $I$ |
|---|---|---|---|---|---|---|
| $10^{-3}$ | $10^{+1}$ | $10^{-1}$ | 15 | 0.33 | $10^{-4}$ | $Identity_{N \times N}$ |

*Table 10.* Hyperparameters used in training for Transformers on IMDb dataset.

We also utilize this implementation for TweetEval, and similar hyperparameters for this sentiment analysis experiment can be found in table 11.

| t | $\tau$ | $\epsilon$ | $\mathcal{S}$ | $\lambda_{\text{initial}}$ | $lr_{initial}$ | $I$ |
|---|---|---|---|---|---|---|
| $10^{-3}$ | $10^{+1}$ | $10^{-1}$ | 15 | 0.33 | $10^{-4}$ | $Identity_{N \times N}$ |

*Table 11.* Hyperparameters used in training for Transformers on TweetEval dataset.

### C.4. Neural Machine Translation

For our neural machine translation experiments, we adopt the Transformer model from `fairseq` along with its DiffTransformer counterpart, training both from scratch for 25 epochs. We then fine-tune them alongside other baselines for an additional 10 epochs on the IWSLT'14 dataset[2]. When fine-tuning ESPFormer and Sinkformer, we modify the original training schedule by reducing the learning rate by a factor of 10.

| t | $\tau$ | $\epsilon$ | $\mathcal{S}$ | $\lambda_{\text{initial}}$ | $lr_{initial}$ | $I$ |
|---|---|---|---|---|---|---|
| $10^{-3}$ | $10^{+1}$ | $10^{-1}$ | 15 | 0.0 | $10^{-4}$ | $Identity_{N \times N}$ |

*Table 12.* Hyperparameters used in training for neural machine translation on IWSLT14 dataset.

### C.5. Vision Transformers

#### C.5.1. CATS AND DOGS CLASSIFICATION

For this experiment, we use the ViT model (Dosovitskiy et al., 2021) with different attention mechanisms. The images are resized to $224 \times 224$. We use a ViT architecture with an embedding and MLP dimension of 128, 6 layers, 8 attention heads, and a patch size of 16. For all methods and percentages of data, we train the model for 300 epochs. We use an initial learning rate of $3 \times 10^{-5}$. After 250 epochs, the learning rate is reduced by a factor of 10. Training is done using the Adam optimizer (Kingma & Ba, 2015) and a batch size of 64. Our experimental setup, including the normalizations and data augmentations, are consistent with the Cats and Dogs experiment in Sinkformer (Sander et al., 2022) . For each percentage of the data, three random seeds are used to initialize and generate new subsets of data. The subsets are consistent across all methods to ensure the same training set is used.

---

[1] https://github.com/lyeoni/nlp-tutorial/tree/master/text-classification-transformer
[2] https://github.com/pytorch/fairseq/blob/main/examples/translation/README.md

| t | $\tau$ | $\epsilon$ | $\mathcal{S}$ | $\lambda_{\text{initial}}$ | $lr_{initial}$ | $I$ |
|---|---|---|---|---|---|---|
| $10^{-3}$ | $10^{-1}$ | 1 | 3 | 0.5 | $3 \times 10^{-5}$ | $Identity_{N \times N}$ |

*Table 13.* Hyperparameters used in training for ViT on Cats and Dogs dataset.

C.5.2. IMPACT OF PATCH SIZE

To analyze the effect of patch size on final accuracy, we conduct experiments using a batch size of 100 and the Adam optimizer. The model architecture consists of a single-layer Transformer (depth = 1) with one attention head, no non-linearity, and varying patch sizes. Training is performed over 45 epochs, with an initial learning rate of $1 \times 10^{-3}$ for the Transformer and DiffTransformer and $2 \times 10^{-3}$ for the ESPFormer and Sinkformer. The learning rate is decayed by a factor of 10 after 35 epochs, and again by another factor of 10 after 41 epochs.

| t | $\tau$ | $\epsilon$ | $\mathcal{S}$ | $\lambda_{\text{initial}}$ | $lr_{initial}$ | $I$ |
|---|---|---|---|---|---|---|
| $10^{-3}$ | 0 | 1 | 5 | 0.5 | $1 \times 10^{-3}, 2 \times 10^{-3}$ | $Identity_{N \times N}$ |

*Table 14.* Hyperparameters used in training for shallow-ViT on MNIST dataset.

