# OpenReview forum: "ESPFormer: Doubly-Stochastic Attention with Expected Sliced Transport Plans"
_ICML.cc/2025/Conference — ICML 2025 poster_

### Official Review · Reviewer_p6Sy · 2025-03-14

**Overall Recommendation:** 2

**Summary:**

This paper introduces ESPFormer, a novel Transformer architecture integrating a fast, doubly-stochastic attention mechanism based on Expected Sliced Transport Plans (ESP). By projecting high-dimensional queries/keys into 1D slices using axis-aligned directions (Θ = I), ESPFormer efficiently computes optimal transport plans via differentiable soft sorting, ensuring end-to-end trainability. The framework aggregates sliced plans using an inverse temperature parameter τ to balance attention sparsity and distribution, achieving O(mN²) runtime with full parallelization across m slices—significantly faster than Sinkhorn’s O(SN²) iterative approach. Experiments demonstrate consistent performance gains across diverse tasks: +0.5% accuracy on ModelNet40 (point clouds), +0.2 BLEU on IWSLT translation, and 6% improvement on Cats&Dogs with 1% data.

**Claims And Evidence:**

The key claims (doubly-stochastic attention, efficiency gains, performance improvements) are supported by experiments across four benchmarks (ModelNet40, IMDb, IWSLT, Cats&Dogs) with consistent metrics (accuracy, BLEU). However, while axis-aligned slices (Θ=I) are justified by parameter efficiency, the paper lacks ablation studies comparing learned vs. fixed slices, leaving uncertainty about their optimality for non-axis-aligned data.

**Essential References Not Discussed:**

No

**Experimental Designs Or Analyses:**

Experiments are generally controlled (same architectures across baselines, multiple runs) but have limitations:
1.No analysis of τ’s impact beyond qualitative visualization (Fig. 2) or slice count (m).
2.While the paper compares ESPFormer with Sinkformer and DiffTransformer on IMDB using accuracy, the narrow focus on a single dataset.

**Methods And Evaluation Criteria:**

The proposed methods (sliced transport, soft sorting, τ-controlled aggregation) are theoretically sound and aligned with ESP’s properties (Liu et al., 2025). Axis-aligned slicing avoids extra parameters but may limit expressiveness for non-axis-separable distributions. Evaluation criteria (accuracy, BLEU, runtime) are standard and relevant.

**Other Comments Or Suggestions:**

no

**Other Strengths And Weaknesses:**

Strengths:
1.Introduces a temperature-controlled sparsity mechanism (via τ), enabling tunable attention patterns without iterative optimization.
2.Parallelizable O(mN²) complexity addresses a critical bottleneck in OT-based attention, making doubly-stochastic constraints viable for long sequences.
Clarity:
3.Well-structured with intuitive visualizations (e.g., Fig. 2: Sinkhorn vs. ESP attention patterns).
4.Appendix details hyperparameters and runtime analysis, aiding reproducibility.

Weaknesses:
1.Axis-aligned slicing is presented as a default choice without justifying why it suffices (vs. learned/adaptive slices). Prior work on adaptive slicing (e.g., [Nguyen et al., 2023]) could strengthen motivation.
2.In the task of image classification, the Cats and Dogs dataset and the MNIST dataset are relatively simple, and there is a lack of verification on more complex datasets.
3.In the Sentiment Analysis experiment, the dataset is single.
4.No analysis of ESPFormer’s scaling to ultra-long sequences (e.g., N=10k), a key use case for efficient attention.

**Questions For Authors:**

1.Why choose axis-aligned slices (Θ=I) over learned or adaptive slicing (e.g., Nguyen et al., 2023)? Did you test adaptive slicing, and if so, how did it compare in terms of accuracy/efficiency?
2.How do τ (inverse temperature) and m (slice count) affect performance? Are optimal τ values consistent across tasks?
3.What the performance when ESPFormer handle ultra-long sequences (N > 10k)? Did you test runtime/accuracy on such data (e.g., text/document classification)?
4.Could you supplement experiments on more diverse datasets? I feel that the improvements in your experimental section are not significant, and the single dataset used may introduce chance factors due to limited diversity.

**Relation To Broader Scientific Literature:**

Key Connections:
Sinkformer (Sander et al., 2022): Directly builds on this work by replacing Sinkhorn iterations with sliced OT for doubly-stochastic attention. The efficiency claim is framed as an improvement over Sinkformer’s O(SN²) complexity.
Sliced OT (Liu et al., 2025): Relies on ESP theory to construct transport plans from 1D slices, adapting it for attention via soft sorting.
Soft Sorting (Prillo & Eisenschlos, 2020): Critical for differentiable sliced transport.

**Theoretical Claims:**

The paper relies on prior work (Liu et al., 2025) for ESP’s theoretical foundation but does not provide new proofs. Assumptions about ESP’s equivalence to Wasserstein distance and τ’s role in sparsity are empirically validated.

---

> ### Author Rebuttal · Authors · 2025-04-01
>
> We appreciate the constructive feedback. Below are our responses.
>
> **1. Axis-aligned slices**
>
> We chose axis-aligned slices because the keys and queries are learned parameters. Thus, any potential optimization of slice orientations is implicitly captured by the query and key matrices, $W_Q$ and $W_K$. This choice enabled us to avoid introducing extra learnable parameters to ensure a fair comparison with baseline methods, such as vanilla attention and Sinkformer.
>
> To investigate the effect of the number and type (learnable vs. frozen) of slices, we conducted additional experiments on the Cats vs. Dogs benchmark, varying the number of slices $L$, the inverse temperature parameter $\tau$, and whether the slices were frozen or learnable. The results are presented below.
>
> | **$\tau=0.1$**   | $L = 1$ | $L = 8$ | $L = 32$ | $L = 64$ | $L = 128$ |
> | ---------------- | ------- | ------- | -------- | -------- | --------- |
> | **Learnable**    | 74.45%  | 78.56%  | 79.33%   | 78.25%   | 76.15%    |
> | **Frozen**       | 66.61%  | 72.75%  | 78.41%   | 79.39%   | 79.66%    |
> | **Axis-Aligned** | —       | —       | —        | 79.47%   | —         |
>
> | **$\tau=1.0$**   | $L = 1$ | $L = 8$ | $L = 32$ | $L = 64$ | $L = 128$ |
> | ---------------- | ------- | ------- | -------- | -------- | --------- |
> | **Learnable**    | 74.45%  | 79.07%  | 78.10%   | 77.64%   | 74.34%    |
> | **Frozen**       | 66.61%  | 73.09%  | 77.89%   | 78.88%   | 78.50%    |
> | **Axis-Aligned** | —       | —       | —        | 78.86%   | —         |
>
> | **$\tau=10$**    | $L = 1$ | $L = 8$ | $L = 32$ | $L = 64$ | $L = 128$ |
> | ---------------- | ------- | ------- | -------- | -------- | --------- |
> | **Learnable**    | 74.45%  | 79.24%  | 78.06%   | 77.07%   | 74.20%    |
> | **Frozen**       | 66.61%  | 73.50%  | 76.79%   | 77.91%   | 78.13%    |
> | **Axis-Aligned** | —       | —       | —        | 77.78%   | —         |
>
> As shown in the table, both learnable and frozen slicers can operate on the key and query projections. We chose a simpler, axis-aligned slicer for interpretability and fair comparison, avoiding extra parameters. Learnable slicers may help with fewer slices by focusing on informative directions, but their advantage diminishes as the slice count increases. In contrast, frozen slicers avoid added complexity but require many slices to capture distributional structure effectively, especially in high dimensions.
>
> **2. Consistency of optimal $\tau$ values across tasks:**
>
> The optimal inverse temperature values, $\tau$, depend on: 1) the input measures, i.e., the task, and 2) the number of slices. Therefore, it is reasonable to expect that this hyperparameter should be task-dependent.  Smaller $\tau$ values promote more diffuse attention, while larger values lead to sharper, more focused (i.e., near one-to-one) attention between tokens. Importantly, as shown in our experimental setup, we perform cross-validation only over a very coarse grid, i.e., $\tau\in\{0.1,1,10,100\}$.
>
> **3. Additional benchmarks:**
>
> We conducted experiments on TweetEval. The table below summarizes the test performance of several models, including ESPFormer, on this benchmark. ESPFormer demonstrates competitive results compared to the baselines, outperforming both Sinkformer and the standard attention mechanism (denoted by "Vanilla"). For a controlled comparison, we replaced the attention module in a standard 6-layer Transformer (as proposed by Vaswani et al., 2017) with alternative components—Sinkformer, DiffAttention, and ESPFormer—while keeping the rest of the architecture unchanged. As illustrated, all variants outperform the vanilla baseline, with ESPFormer and DiffAttention achieving the highest accuracies on this benchmark.
>
>
> | Model    | Vanilla | Sink. | Diff.Att. | ESP. |
> | -------- | ------- | ----- | --------- | ---- |
> | Accuracy | 71.5    | 72.0  | 72.6      | 72.6 |
>
> Note that this evaluation is limited to a single architecture and run due to the rebuttal timeframe. We plan to extend it in the camera-ready version with plug-and-play experiments across diverse attention modules and pretrained language models.
>
> **4.  ESPFormer on ultra-long sequences (N > 10k)**
>
> Due to time and resource constraints, we couldn't evaluate ESPFormer on ultra-long sequences (N > 10k) in this submission. We agree this is important and plan to include full runtime and accuracy benchmarks in future work. Regarding efficiency, by switching from soft-sorting to hard sorting during inference, we can significantly reduce our latency.
>
> Once again, we sincerely thank the reviewer for their thoughtful comments and insightful questions, which helped us refine the presentation and analysis of our contributions.
>
> **P.S.** Due to the limited rebuttal period, the results presented are based on a single run. We recognize the importance of reporting performance across multiple seeds and will include averaged results in the camera-ready version.

---

### Official Review · Reviewer_ZJyu · 2025-03-14

**Overall Recommendation:** 4

**Summary:**

This paper is about achieving doubly stochastic attention in Transformers through ESP (expected sliced transport plans). The softmax is known to be a notorious bottleneck for the expressivity and gradient flow in Transformers, thus is clearly a relevant research area. Moreover, prior art like the Sinkformer rely on iterative approximation procedures to obtain DSMs. The authors ensure differentiability of their ESP approach and and show empirically that performance improves across multiple datasets. Moreover, the authors introduce a flexible hyperparameter (tao) which allows to control the temperature of the attention while ensuring doubly stochasticity.

## update after rebuttal:
I had originally given a score of 4 to this paper due to high methodological novelty, sound experimental design and clear relevance for the field. The score was contingent on the author's response to some raised concerns and questions, which were adequately addressed. I keep my score.

**Claims And Evidence:**

The general claims of the paper are well supported by experimental evidence.
- The ESPFormer ensures a more balanced distribution (than Softmax) and gives flexibility via inverse temperature $\tau$.
- Performance improvement over Transformer (and even Sinkformer and DiffFormer) is clear in all cases, even though the hyperparameter sweeps to reach these results are not made explicit, so it is unclear how the authors arrived e.g., at different values of $\tau$.
- ESPFormer can work as a drop-in replacement for models trained with standard attention of DiffAttention and can still boost performance during finetuning.

Overall, I think that this is an exceptional  and very innovative paper. I still have some concerns that require clarification but I believe that this is a very valuable piece of research that will help shaping the future of attention and Transformers.

**Essential References Not Discussed:**

The authors did a great job at revising prior literature about adaptations of Transformers, especially toward doubly stochastic attention. An alternative potential avenue for doubly stochastic Transformer could be quantum computing as was highlighted in the conclusion of this paper from ICML last year: https://proceedings.mlr.press/v235/mariella24a.html

**Experimental Designs Or Analyses:**

The experimental results are convincing and well designed, just ablation study and interpretations on the impact of $\tau$ are missing, ideally on more than one dataset.

Fig 2: Effectively, the choice of tao seems to interpolate between the extremes of tao=100 where the attention matrix is a permutation matrix and tao=0 where the DSM is almost the identity matrix -- in other words we seem to move between the vertices and the origin of the Birkhoff polytope. Can you confirm (or even prove) this intuition? If yes, it migh be worth emphasizing this in the text to ease interpretability of the parameter tao.

**Methods And Evaluation Criteria:**

The authors empirically evaluate ESPFormers across a rich set of domain and show convincing results everywhere. So in general, evaluation criteria are absolutely convincing.


However, at least one ablation study on the impact of $\tau$ is necessary for the paper. From the Appendix, I understand that the authors tuned tau specifically for each dataset, yet they are not transparent on the conducted hyperparameter sweeps, raising some questions on how many attempts the ESPFormer was given to obtain the results finally shown in the paper.

**Other Comments Or Suggestions:**

- Fig 2: Effectively, the choice of tao seems to interpolate between the extremes of tao=100 where the attention matrix is a permutation matrix and tao=0 where the DSM is almost the identity matrix -- in other words we seem to move between the vertices and the origin of the Birkhoff polytope. Can you confirm (or even prove) this intuition? If yes, it migh be worth emphasizing this in the text to ease interpretability of the parameter tao.

- Are your DSMs exact or approximate? In the Sinkformer they are only approximate due to the iterative nature of Sinkhorn's Algorithm.

- For Figure 3, for completeness S=3 should be added since it is empirically often enough as shown in the Sinkformer.

Section 3.3:
- Please mention for completeness also the complexity of the standard attention following the same notation
- Whether your complexity is better than the Sinkformer, in theory, depends on (1) whether m < S and (2) d < N. Regarding (1): Typicall no because the Sinkformer shows good performance with S being as small as 3 or 5 and observes saturation around S=20. Regarding (2): typically yes, your d should correspond to d_v in the standard Transformer). Even if this is more of a theoretical than a practical perspective, I feel amplifying it would improve clarity



Minor:
L412 (right side) - missing space

**Other Strengths And Weaknesses:**

Positive:
- Strong theoretical motivation, clear novelty
- Strong empirical results at only mild runtime increase

Negative:
- No source code is provided, hampering reproducibility
- A small discussion on when to chose which tao would be helpful

**Questions For Authors:**

-

**Relation To Broader Scientific Literature:**

This paper suggests a clear improvement over the Sinkformer which has itself shown to improve over standard attention in Transformer. It removes the limitation of Sinkformer that only produces DSMs as iterative approximations and shows superior empirical results.

**Theoretical Claims:**

(1) FIg1 suggests (visually speaking) that the slice-specific DSMs are actually permutation matrices rather than interior points of the Birkhoff polytope. Compare eq. (8), this seems to originate from the deltas being diracs (BTW, this is never said explicitly!), yet in Eq (8) neither the sigmas nor the Us seem to be binary to me. Please clarify! Moreover, if the slices of G are indeed permutation matrices, maybe you want to emphasize the Birkhoff von Neumann theorem which is that every DSM (every element in the birkhoff polytope) can be expressed as a convex combination of permutation matrices. This brings me to another question about empirical and theoretical expressivity. Can you guarantee that every DSM can be reached with your algorithm? The BvN theorem requires non-negative coefficients theta which are more flexible then the simple averaging you do to obtain the ESP Attention matrix. Ultimately, may the ESPFormer be complemented with an extra set of L parameters enforced to sum to 1 that allow a weighted average of the slice-specific DSMs in order to obtain full expressivity over the Birkhoff polytope?

(2)
Whether the runtime of $O(LN logn)$  for transport plan computation is lower than $O(N^2 logN)$ depends on the number of slices. Since you avoid adding extra parameters (which I agree is a good choice), you set $L=m$ which is called $d_k$ in standard attention. Indeed, typically $d_k << N$ in transformers in practice, still it should be emphasized that we are not talking about a theoretical speedup, just about a practical one, due to common design choices when deciding about transformer hyperparameters. Nobody prevents me from setting $d_k > N$ and in that case, IIUC, the runtime complexity would be lower.

(3) Section 3.2 -- Dimensionalities requires clarification. It seems there is a mixup on the matrix dimension because they are swapped? I recommend to follow the notation from the original paper by Vaswani et al, where we start with X of size $N x d$. Instead you seem to start from X of shape $d x N$, so you are left-multiplying the inputs with the $WQ$ matrix to obtain Q and K of shape m x N. Standard attention runs $QK^T$ to get a matrix of shape $NxN$ whereas in your case you have to compute $K^TQ$ to get a $NxN$ matrix which is very unconventional. To ease readability, I suggest to update all notation. For the same reason in (12) you arrive at $VG$ rather than $GV$.
Also W_V needs to be quadratic according to your definition, whereas in general it does not have this constraint. Please clarify

---

> ### Author Rebuttal · Authors · 2025-04-01
>
> Thank you for the constructive feedback. Below, we provide our responses.
>
> **1. The Birkhoff Polytope.**
> We thank the reviewer for raising this insightful question.
> In the context of the Birkhoff polytope $B_N$, we will consider the transport plans between $\mu=\sum_{i=1}^N\delta_{x_i}$ and $\nu=\sum_{j=1}^N\delta_{y_j}$, i.e., two measures having the same number of supports $N$ and uniformly-distributed mass. To avoid confusion, we refer to the "identity matrix" in $B_N$ as the outer product plan and denote the $N\times N$ matrix with all entries equal to $1$ as $\mathbb{1}_{N\times N}$. The following fact about $B_N$ is used in the following analysis [1]:
> * (Birkhoff-von Neumann Theorem) $B_N$ is a convex polytope whose extreme points, or corners, are the $N!$ permutation matrices.
> 1. When $\tau\rightarrow\infty$, a permutation matrix is recovered. In the original EST work [2], the authors have mentioned that the distribution of $\theta$ will degenerate into a one-hot vector as $\tau\rightarrow\infty$. The EST will recover min-SWGG [3], where the minimum is taken over all permutation matrices induced by the slices, resulting in a permutation matrix. By the Birkhoff-von Neumann Theorem, this will be one of the corners of $B_N$.
> 2. When $\tau=0$, the EST is not at the origin, i.e., the outer product measure $\frac{1}{N^2}\mathbb{1}_{N\times N}$. In Figure 6 of [2], EST with varying $\tau$ is compared with the Optimal Transport plan, the entropic Optimal Transport plan, and the outer product plan. EST is evidently different from the outer product measure. It is worth mentioning that entropic Optimal Transport plan interpolates between the outer product and the Optimal Transport plan, whereas EST does not.
> 3. In [this figure](https://github.com/Anon4142/ESPFormer/blob/main/figures/dist_perm.png), we provide illustration for $\tau=0$ and $\tau\rightarrow \infty$ using a pair of distributions $\mu, \nu$, each supported on 3 points. The corners of $B_3$ consists of $S_0, S_1, \cdots, S_5$. The pie chart in this figure shows the optimal transport plan (i.e., the optimal permutation) as a function of the slicing angle. As can be seen $S_1$ and $S_3$ are the highly voted transportation plans. When $\tau=0$, the EST is a convex combination of the corners with weights indicated in the pie chart.  When $\tau\rightarrow \infty$, the EST will recover the one permutation that gives the minimal transport cost, which is $S_1$ in this example.
>
> [1] Birkhoff, G., 1946.
> [2] Liu X, et al. 2024.
> [3] Mahey G, et al. 2023.
>
> **2. What $\tau$ to choose?**
> The optimal value of the hyperparameter $\tau$ depends on the input probability measures (i.e., the task) and the number of slices. Smaller $\tau$ values promote more diffuse attention, while larger values lead to sharper, more focused (i.e., near one-to-one) attention between tokens. We ran more ablative studies on the choice of $\tau$ and slicing strategies. Please see responses to reviewer eE5t.
>
> **3. Source code availability.**
> We apologize for the unclear placement of the code link, which was included as a footnote on page 7. In the camera-ready version, we will move it to the abstract for better visibility. Meanwhile, the anonymous GitHub repository is available [here](https://github.com/Anon4142/ESPFormer).
>
> **4. Exact/Approximate DSMs.**
> Soft sorting yields approximate DSMs during training, but annealing the temperature gradually transitions it toward hard sorting and exact DSMs. At inference, hard sorting can replace soft sorting, reducing complexity from $O(N^2)$ to $O(LN\log N)$. We will include this discussion in the revised paper.
>
> To demonstrate, we fine-tuned ESPFormer for 40 epochs on Cats vs. Dogs (1% and 10% data) and replaced soft sorting with hard sorting at inference. Results show improved accuracy and efficiency:
>
> | Data Fraction | Initial (Soft) | After Annealing | Hard Sort |
> |---------------|----------------|------------------|-----------|
> | 1%            | 59.87%         | 60.50%           | 61.02%    |
> | 10%           | 71.71%         | 72.16%           | 72.66%    |
>
> Experiments on 25% and 100% data are ongoing and will be included in the camera-ready version.
>
> **5. Updating Figure 3.**
> We have updated Figure 3 to include the runtime for the Sinkhorn method with $S=3$. We also added the inference runtime for ESP via hardsorting (i.e., inference time). The revised Figure 3 is available [here](https://github.com/Anon4142/ESPFormer/blob/main/figures/esp_sink_runtime.png).
>
> **6. On the complexity of ESP, Sinkhorn and Vanilla Attention.**
> We observe that with SoftSort, ESP achieves lower runtime complexity than Sinkhorn for $S \geq 5$, though it remains slower when $S = 3$ (for training). While $S = 3$ or $5$ may be sufficient for some Sinkformer tasks, others, such as ModelNet40, require more iterations (e.g., $S = 21$). We will clarify this and include the complexity of vanilla attention in the main manuscript.
>
> Thank you again for your consideration.

---

### Official Review · Reviewer_eE5t · 2025-03-17

**Overall Recommendation:** 4

**Summary:**

The paper introduces ESPFormer, a attention mechanism that enforces doubly-stochastic constraints in attention matrices without requiring iterative Sinkhorn normalization. Instead, it leverages Expected Sliced Transport Plans (ESP) to achieve a fully parallelizable and computationally efficient solution. The authors integrate a temperature-based soft sorting technique to ensure differentiability, making ESPFormer compatible with deep learning models. Experimental evaluations across diverse applications, including image classification, point cloud classification, sentiment analysis, and neural machine translation, demonstrate the advantages of ESPFormer over traditional self-attention and Sinkhorn-based attention methods.

## update after rebuttal

I increased the score

**Claims And Evidence:**

1. ESPFormer provides a more computationally efficient alternative to Sinkhorn-based doubly-stochastic attention mechanisms.

2. The proposed attention mechanism leads to better-balanced attention distributions, improving model performance.

3. ESPFormer enhances downstream tasks such as classification and translation compared to classical self-attention and Sinkhorn-based approaches.

4. Fine-tuning pre-trained models with ESPFormer leads to significant performance improvements.

These claims are supported by experimental evidence on multiple datasets. The runtime complexity analysis demonstrates computational advantages, and performance metrics from classification and translation tasks confirm improved accuracy.

**Essential References Not Discussed:**

While the paper discusses relevant prior work, additional references on non-iterative attention normalization techniques or alternative soft sorting methods might provide further context.

**Experimental Designs Or Analyses:**

The experiments are well-structured, covering a range of tasks that benefit from improved attention mechanisms. The paper provides:

1. Comparisons against baseline models (Vanilla Transformer, Sinkformer, DiffTransformer) across various datasets.

2. Analysis of computational efficiency via runtime evaluations.

3. Fine-tuning experiments to validate the plug-and-play nature of ESPFormer.

**Methods And Evaluation Criteria:**

The methods proposed are well-motivated for addressing the limitations of standard self-attention and Sinkhorn normalization. The evaluation criteria include:

1. Accuracy on benchmark datasets for classification tasks.

2. BLEU scores for machine translation tasks.

3. Runtime complexity comparisons between ESPFormer and Sinkformer.

4. Ablation studies on hyperparameters such as inverse temperature.

These metrics appropriately reflect the benefits of the proposed method in terms of both efficiency and performance.

**Other Comments Or Suggestions:**

Since language model  is one of the most important application of Transformer, experiments on  language model (e.g., on WikiText-103) should be considered.

**Other Strengths And Weaknesses:**

## Strengths

1.  Introduces a novel, computationally efficient alternative to Sinkhorn-based attention mechanisms, eliminating the need for iterative normalization.

2. The method is fully parallelizable, making it more scalable than Sinkhorn-based approaches, which require sequential iterations.

3. Strong theoretical grounding connects ESP attention with optimal transport theory, ensuring the approach is well-principled.

4. Extensive experiments demonstrate the effectiveness of ESPFormer across a variety of tasks, including image classification, point cloud classification, sentiment analysis, and neural machine translation.

5. The paper provides clear empirical evidence that replacing self-attention mechanisms in pre-trained models with ESPFormer leads to performance improvements with minimal fine-tuning.

6. The introduction of the inverse temperature hyperparameter allows fine-grained control over attention sparsity, adding adaptability across different tasks.

## Weaknesses

1. The reliance on soft sorting introduces an additional hyperparameter (temperature), which may require careful tuning for different tasks.

2. Additional experiments on language model is needed to demonstrate further the performance of ESPFormer.

**Questions For Authors:**

1. How does ESPFormer handle very long sequences compared to efficient self-attention variants like Linformer? Is it possible to reduce further the complexity of ESPFormer?

2. I understand that the choice of Identity matrix is for computational convenience since it does not require projection. However, it is worth to try if a fixed set of directions $\theta_1,\ldots,\theta_L$ which are not the standard basis can affect the performance (L can be larger than d). This set of directions can be randomly sampled from the uniform distribution or using quasi Monte Carlo methods to encourage repulsive structure [1].

[1] Quasi-Monte Carlo for 3D Sliced Wasserstein, Nguyen et al.

**Relation To Broader Scientific Literature:**

ESPFormer extends prior work on optimal transport-based attention mechanisms by addressing computational bottlenecks and using sliced optimal transport plans.

**Theoretical Claims:**

N/A

---

> ### Author Rebuttal · Authors · 2025-04-01
>
> We thank the reviewer for their constructive feedback. Below are our responses.
>
> **1. Reliance on Soft sorting:**
>
> Soft sorting introduces a temperature hyperparameter that requires tuning. To address this, we use *temperature annealing*, an exponential decay schedule, to smoothly transition from soft to hard sorting. This allows the model to adapt as sorting sharpens, enabling hard sorting at inference without performance loss.
> This improves runtime, reducing complexity from \( O(N^2) \) to \( O(N \log N) \), and reduces our dependency on soft-sorting. See [Figure 3](https://github.com/Anon4142/ESPFormer/blob/main/figures/esp_sink_runtime.png) for updated runtimes. The table below summarizes results on Cats vs. Dogs across training fractions.
>
> | Data Fraction | Initial Test Acc (Soft Sort) | After Temp Decay (Sharp Soft Sort) | After Switching to Hard Sort |
> |---------------|------------------------------|-------------------------------------|-------------------------------|
> | 1%            | 59.87%                        | 60.50%                              | 61.02%                        |
> | 10%           | 71.71%                        | 72.16%                              | 72.66%                        |
>
> Experiments for the 25% and 100% training splits are currently in progress and will be included in the **camera-ready version** of the paper.
>
>
> **2. Additional experiments:**
>
> We conducted experiments on TweetEval. The table below summarizes the test performance of several models, including ESPFormer, on this benchmark. ESPFormer demonstrates competitive results compared to the baselines, outperforming both Sinkformer and the standard attention mechanism (denoted by "Vanilla"). For a controlled comparison, we replaced the attention module in a standard 6-layer Transformer (as proposed by Vaswani et al., 2017) with alternative components—Sinkformer, DiffAttention, and ESPFormer—while keeping the rest of the architecture unchanged. As illustrated, all variants outperform the vanilla baseline, with ESPFormer and DiffAttention achieving the highest accuracies on this benchmark.
>
>
> | Model    | Vanilla | Sink. | Diff.Att. | ESP. |
> | -------- | ------- | ----- | --------- | ---- |
> | Accuracy | 71.5    | 72.0  | 72.6      | 72.6 |
>
> Note that this evaluation is limited to a single architecture and run due to the rebuttal timeframe. We plan to extend it in the camera-ready version with plug-and-play experiments across diverse attention modules and pretrained language models.
>
>
> **3. Axis-aligned slices:**
>
> We chose axis-aligned slices because the keys and queries are learned parameters. Thus, any potential optimization of slice orientations is implicitly captured by the query and key matrices, $W_Q$ and $W_K$. This choice enabled us to avoid introducing extra learnable parameters to ensure a fair comparison with baseline methods, such as vanilla attention and Sinkformer.
>
> To investigate the effect of the number and type (learnable vs. frozen) of slices, we conducted additional experiments on the Cats vs. Dogs benchmark, varying the number of slices $L$, the inverse temperature parameter $\tau$, and whether the slices were frozen or learnable. The results are presented below.
>
> | **$\tau=0.1$**   | $L = 1$ | $L = 8$ | $L = 32$ | $L = 64$ | $L = 128$ |
> | ---------------- | ------- | ------- | -------- | -------- | --------- |
> | **Learnable**    | 74.45%  | 78.56%  | 79.33%   | 78.25%   | 76.15%    |
> | **Frozen**       | 66.61%  | 72.75%  | 78.41%   | 79.39%   | 79.66%    |
> | **Axis-Aligned** | —       | —       | —        | 79.47%   | —         |
>
> | **$\tau=1.0$**   | $L = 1$ | $L = 8$ | $L = 32$ | $L = 64$ | $L = 128$ |
> | ---------------- | ------- | ------- | -------- | -------- | --------- |
> | **Learnable**    | 74.45%  | 79.07%  | 78.10%   | 77.64%   | 74.34%    |
> | **Frozen**       | 66.61%  | 73.09%  | 77.89%   | 78.88%   | 78.50%    |
> | **Axis-Aligned** | —       | —       | —        | 78.86%   | —         |
>
> | **$\tau=10$**    | $L = 1$ | $L = 8$ | $L = 32$ | $L = 64$ | $L = 128$ |
> | ---------------- | ------- | ------- | -------- | -------- | --------- |
> | **Learnable**    | 74.45%  | 79.24%  | 78.06%   | 77.07%   | 74.20%    |
> | **Frozen**       | 66.61%  | 73.50%  | 76.79%   | 77.91%   | 78.13%    |
> | **Axis-Aligned** | —       | —       | —        | 77.78%   | —         |
>
> Learnable slicers may help with fewer slices by focusing on informative directions, but their advantage diminishes as the slice count increases. In contrast, frozen slicers avoid added complexity but require many slices to capture distributional structure effectively, especially in high dimensions.
>
> Once again, we thank the reviewer for their time and consideration,.
>
>  **P.S.** Due to the limited rebuttal period, the results presented are based on a single run. We recognize the importance of reporting performance across multiple seeds and will include averaged results in the camera-ready version.

---

> > ### Comment · Reviewer_eE5t · 2025-04-01
> >
> > I would like to thank the authors for a detailed rebuttal and additional ablation studies. I raised my score to 4 since my questions were answered.  I believe this work is a great connection between the sliced optimal transport literature and deep learning architecture literature. I suggest the author include more discussion on related works (as suggested by reviewers) in both literature to form a strong connection.

---

> > > ### Author Response · Authors · 2025-04-01
> > >
> > > We sincerely thank the reviewer for their thoughtful feedback and for raising their score.
> > >
> > > We will ensure that all missing references are included in the camera-ready version of the paper and add the suggested discussions on related works.

---

### Decision · Program_Chairs · 2025-05-01

**Decision:**

Accept (poster)

**Comment:**

In this study, the authors propose a new variant of Transformer, called ESPFormer, which achieves doubly stochastic attention maps via simple operations like sorting. The authors analyze and interpret the model from the perspective of sliced optimal transport, and compare the model with the existing OT-based Transformer (Sinkformer) on the datasets. Experimental results show the feasibility of the proposed method to some extent.

AC and reviewers admit the technical novelty of the proposed method. However, some highly relevant references [1, 2], although preprints, should be added as related work, and the discussion and comparison with these works should be added in the paper. In addition, AC believes that the experimental part should be enhanced because the datasets and the tasks considered in the paper are relatively simple. In summary, AC recommends accepting this work weakly.

[1] Yuan, S., & Xu, H. (2024). Towards Better Multi-head Attention via Channel-wise Sample Permutation. arXiv preprint arXiv:2410.10914.

[2] Yuan, S., & Xu, H. (2023). Sliceformer: Make Multi-head Attention as Simple as Sorting in Discriminative Tasks. arXiv preprint arXiv:2310.17683.